# Assessment of Antarctic Sea Ice Cover in CMIP6 Prediction with Comparison to AMSR2 during 2015–2021

**Siqi Li** [1], **Yu Zhang** [1,2,*] , **Changsheng Chen** [3], **Yiran Zhang** [1], **Danya Xu** [2] and **Song Hu** [1]

1 College of Marine Sciences, Shanghai Ocean University, Shanghai 201306, China; shu@shou.edu.cn (S.H.)
2 Southern Marine Science and Engineering Guangdong Laboratory (Zhuhai), Zhuhai 519082, China
3 School for Marine Science and Technology, University of Massachusetts-Dartmouth, New Bedford, MA 02744, USA
* Correspondence: yuzhang@shou.edu.cn; Tel.: +86-021-6190-0342

**Abstract:** A comprehensive assessment of Antarctic sea ice cover prediction is conducted for twelve CMIP6 models under the scenario of SSP2-4.5, with a comparison to the observed data from the Advanced Microwave Scanning Radiometer 2 (AMSR2) during 2015–2021. In the quantitative evaluation of sea ice extent (SIE) and sea ice area (SIA), most CMIP6 models show reasonable variation and relatively small differences compared to AMSR2. CMCC-CM4-SR5 shows the highest correlation coefficient (0.98 and 0.98) and the lowest RMSD ($0.98 \times 10^6$ km$^2$ and $1.07 \times 10^6$ km$^2$) for SIE and SIA, respectively. In the subregions, the models with the highest correlation coefficient and the lowest RMSD for SIE and SIA are inconsistent. Most models tend to predict smaller SIE and SIA compared to the observational data. GFDL-CM4 and FGOALS-g3 show the smallest mean bias ($-4.50$ and $-1.21 \times 10^5$ km$^2$) and the most reasonable interannual agreement of SIE and SIA with AMSR2, respectively. In the assessment of sea ice concentration (SIC), while most models can accurately predict the distribution of large SIC surrounding the Antarctic coastal regions, they tend to underestimate SIC and are unable to replicate the major patterns in the sea ice edge region. GFDL-CM4 and FIO-ESM-2-0 exhibit superior performance, with less bias (less than $-5\%$) and RMSD (less than 23%) for SIC in the Antarctic. GFDL-CM4, FIO-ESM-2-0, and CESM2 exhibit relatively high positive correlation coefficients exceeding 0.60 with the observational data, while few models achieve satisfactory linear trend prediction of SIC. Through the comparison with RMSD, Taylor score (TS) consistently evaluates the Antarctic sea ice cover and proves to be a representative statistical indicator and applicable for its assessment. Based on comprehensive assessments of sea ice cover, CESM2, CMCC-CM4-SR5, FGOALS-g3, FIO-ESM-2-0, and GFDL-CM4 demonstrate more reasonable prediction performance. The assessment findings enhance the understanding of the uncertainties associated with sea ice in the CMIP6 models and highlighting the need for a meticulous selection of the multimodel ensemble.

**Keywords:** Antarctic; CMIP6; AMSR2; sea ice extent; sea ice area; sea ice concentration

## 1. Introduction

The sea ice is a critical component of the Earth's climate system and exerts a significant influence on the exchange of gas, heat, and momentum between the high-latitude oceans and the atmosphere [1–3]. In addition, the formation and melting of sea ice have a remarkable impact on the generation and propagation of ocean waves due to the air–sea-ice interaction [4,5]. Sea ice concentration (SIC) is one of the most key parameters of sea ice, and it guides the estimation of sea ice extent (SIE), and sea ice area (SIA). Under the background of global warming, the Antarctic sea ice cover has unexpectedly not declined as precipitous as Arctic sea ice cover [6,7]. Actually, the total Antarctic SIE has been revealed as a slight increase of approximately $1.7 \pm 0.2\%\cdot$decade$^{-1}$ during 1979–2014 [8]. The Bellingshausen and Amundsen Sea exhibited a negative trend of SIE at $22.5 \pm 0.7\%\cdot$decade$^{-1}$ in spite of

the positive trends observed in all other sectors from 1979 to 2015 [8,9]. The SIE in the total Southern Ocean gradually culminated in 2014 with the annual mean of $12.8 \times 10^6$ km$^2$, followed by a sudden drop that reduced to a historic low of $10.7 \times 10^6$ km$^2$ for annual mean in 2017. This significant loss of sea ice cover during 2014–2017 is equivalent to a 30-year decrease in Arctic sea ice [10–12]. After that, the Antarctic sea ice cover did not recover and eight new lowest monthly records appeared during 2016–2020 [13]. The uneven changes in Antarctic sea ice potentially cause significant impacts and disaster risks to the polar climates and ecosystems [14,15]. Moreover, the particular change presents unprecedented challenges in understanding the future variations of Antarctic sea ice and the associated climate change. Improved precision and reliability in sea ice predictions are essential for enhancing the comprehension of the responses of sea ice variability to climate change. Presently, future prediction of Antarctic sea ice is predominantly reliant on diverse numerical models [16–18]. The primary objective of the coupled model intercomparison project (CMIP) is to explore the intricate processes of the Earth system subjected to climate forcing and assess the efficacy of climate models in predicting future trends. In recent years, the implementation of CMIP Phase 6 (CMIP6) has led to the release of updated models that encompass sea ice simulations. Shu et al. [19] made a comparison of CMIP6 simulation of SIE with CMIP5 simulations and satellite observation and found that the multimodel mean of CMIP6 can adequately reproduce the seasonal cycles of Antarctic SIE. Analogous to the limitations of the majority of CMIP5 models in reproducing the observed positive trend of Antarctic SIE over the period 1979–2005 [20,21], the CMIP6 multimodel ensemble also exhibits a similar inadequacy in capturing this trend. Nevertheless, it is worth noting that the intermodel spread within CMIP6 has exhibited a reduction in comparison to those in CMIP5. Despite significant negative trends in SIA observed in many CMIP6 models over the period 1979–2018, they are capable of capturing the main characteristics of the mean seasonal cycle of SIA. Moreover, the intermodel spread in SIA has decreased [22]. The investigation of high-frequency variability in SIE reveals a reduced level of bias compared to the observation as well as a large spread in variability across CMIP6 models [23].

However, the studies of Antarctic sea ice cover above primarily involve historical simulations of CMIP6 and the multimodel ensemble. Few studies have focused on the assessment of CMIP6 sea ice cover predictions and the detailed differences among climate models. Since prediction datasets of CMIP6 started from 2015, observed sea ice parameters derived from the satellites provide the possibility to validate the results of recent years. Therefore, to assess the capability of CMIP6 in projecting the spatio-temporal variation of Antarctic SIE, SIA, and SIC in recent years, we have selected twelve models from different institutions across eight countries. Through assessment with satellite observation from 2015 to 2021, we aim to identify models that are more credible for future predictions of Antarctic sea ice. Rather than relying on the multimodel ensemble, we quantify differences between each model and observed data, examining monthly, seasonal, and annual variations in different subregions of the Antarctic Ocean.

## 2. Data and Methods

### 2.1. CMIP6 Prediction Data

In this study, we have employed Antarctic SIC data from CMIP6 under the intermediate greenhouse gas emissions scenario of SSP2-4.5 as the prediction datasets [24]. Based on the collected model datasets, twelve different models from research institutions in eight countries are selected. The monthly mean SIC data cover the period from 2015 to 2021. More detailed information about each of the climate models used in this study is listed in Table 1.

**Table 1.** The information of CMIP6 models used.

| Model Name | Country (Institution) | Sea Ice Component | Resolution (Lon × Lat) |
|---|---|---|---|
| CESM2 | United States (NCAR) | CICE5.1 | 320 × 384 |
| CIESM | China (THU) | CICE4.1 | 320 × 384 |
| CMCC-CM2-SR5 | Italy (CMCC) | CICE4.0 | 360 × 291 |
| EC-Earth3-CC | Europe (EC) | NEMO-LIM3 | 362 × 292 |
| FGOALS-g3 | China (CAS) | CICE4.0 | 360 × 218 |
| FIO-ESM-2-0 | China (FIO) | CICE4.0 | 320 × 384 |
| GFDL-CM4 | United States (NOAA/GFDL) | GFDL-SIM4p25 | 1440 × 1080 |
| MIROC-ES2L | Japan (MIROC) | COCO4.9 | 360 × 256 |
| MPI-ESM1-2-HR | Germany (MPI-M) | Unnamed | 802 × 404 |
| NESM3 | China (NUIST) | CICE4.1 | 320 × 384 |
| NorESM2-MM | Norway (NCC) | CICE | 360 × 384 |
| UKESM1-0-LL | United Kingdom (NCAS and MOHC) | CICE-HadGEM3-GSI8 | 360 × 330 |

*2.2. Observed Data*

The satellite observed data of SIC are obtained from Advanced Microwave Scanning Radiometer 2 (AMSR2), which is available through the National Snow and Ice Data Center [25]. AMSR2 is widely used in the research field of sea ice due to its high resolution of SIC [26,27]. The spatial resolution used in this study is 12.5 km × 12.5 km. The monthly SIC data were derived by computing the average of daily SIC data during 2015–2021.

To account for the difference in grid resolution between the CMIP6 climate models and satellite observation, it is necessary to resample the data to a consistent grid size. In this study, we employed the inverse distance weighting (IDW) interpolation technique to convert the model data grids into observed data grids with the resolution of 12.5 km × 12.5 km. The SIE was derived by computing the total area of grid-cells with a threshold of sea ice concentration exceeding 15%. By maintaining the same concentration threshold, the SIA was derived by multiplying SIC with individual grid-cell areas, before summing up the total area.

*2.3. Division of Subregions*

The research area encompasses the Southern Ocean with latitude greater than 55°S. Due to oceanic and climatic similarities, the Southern Ocean can be subdivided into five subregions including the Weddell Sea (60°W~20°E), Indian Ocean (20°E~90°E), Western Pacific Ocean (90°E~160°E), Ross Sea (160°E~130°W), and Bellingshausen and Amundsen (B & A) Seas (130°W~60°W), as depicted in Figure 1 [28].

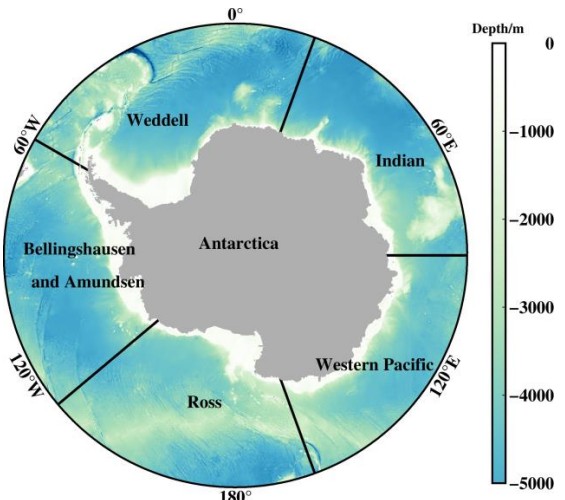

**Figure 1.** Map of the Southern Ocean and five subregions.

### 2.4. Comprehensive Assessment Methods

In order to evaluate the performance of models in CMIP6, statistical indicators were employed, including correlation coefficient, bias (model minus observation), root mean square difference (RMSD), median value, inter-quartile range, and linear trend analysis. Additionally, the Taylor score (TS) was used [29], and the comparison between TS and traditional statistical indicators was discussed. TS is associated with the combination of correlation coefficient and standard deviation which is defined as

$$\text{TS} = \frac{4(1 + R)^4}{\left( \frac{\sigma_{sm}}{\sigma_{so}} + \frac{\sigma_{so}}{\sigma_{sm}} \right)^2 (1 + R_0)^4}, \tag{1}$$

where $R$ is the correlation coefficient between observed and simulated data, and $R_0$ is the maximum correlation coefficient of all models. $\sigma_{sm}$ and $\sigma_{so}$ are the standard deviation of models and observation, respectively. The range of *TS* is from 0 to 1. The value of *TS* closer to 1 indicates the better performance of the model.

## 3. Results

### 3.1. The Assessment of SIE and SIA Predictions

#### 3.1.1. Monthly Variation and Regional Differences

At first, we evaluated the monthly variation of SIE and SIA across the Antarctic region and five subregions during 2015–2021 (Figure 2). The results indicate that most CMIP6 models have the capability to predict the reasonable monthly variation of SIE. The correlation between most CMIP6 models and the observation is relatively high with correlation coefficients larger than 0.65. CMCC-CM4-SR5 exhibits the largest coefficients of 0.98 and 0.96 in the Antarctic and Indian Ocean, respectively. NESM3, NorESM2-MM, GFDL-CM4, and CIESM show the highest correlations (R = 0.96, 0.95, 0.91, and 0.80), respectively, in the Weddell Sea, Western Pacific Ocean, Ross Sea, and B & A Seas. The observed SIE estimated using AMSR2 generally displays minima in February and maxima in September throughout the Antarctic domain (Figure 2). Among the twelve CMIP6 models, only three models (CMCC-CM2-SR5, MIROC-ES2L, and NorESM2-MM) can capture these characteristics. In the subregions, the minima appears between February and March, and the maxima between August and October. Only CIESM, NorESM2-MM, and MPI-ESM1-2-HR exhibit consistent agreement with these temporal patterns across all subregions. A few models fail to capture the characteristics in some years. For example, FGOALS-g3, UKESM1-0-LL, and CESM2 are unable to match the time periods of maxima in the Weddell Sea, Western Pacific Ocean, and B & A Seas, respectively. In addition, these three models fail to match the maxima in the Ross Sea. CMCC-CM2-SR5, EC-Earth3-CC, and MIROC-ES2L have relatively weak performance to match the time periods of minima.

Compared with SIE, the CMIP6 models show a relatively high correlation of SIA with the observation. Most models have correlation coefficients greater than 0.70 in the Antarctic Ocean and all the subregions. As estimated above, CMCC-CM4-SR5 still has the highest correlation coefficients of 0.98 and 0.96 among all 12 models in the Antarctic and Indian Ocean. NESM3, NorESM2-MM, FIO-ESM-2-0, and UKESM1-0-LL show the highest coefficient (R = 0.95, 0.94, 0.94, and 0.82) in the Weddell Sea, Western Pacific Ocean, Ross Sea, and B & A Seas, respectively. The observed SIA generally shows the similar time periods of minima and maxima as SIE in the Antarctic and subregions; however, none of the models can accurately capture the characteristics in all regions (Figure 2). The number of models that failed to capture maxima in SIA is 1, 4, 1, 0, and 8 in the Weddell Sea, Indian Ocean, Western Pacific Ocean, Ross Sea, and B & A Seas, respectively. The number of models that did not match the time period of minima in SIA becomes 7, 6, 7, 5, and 1, respectively.

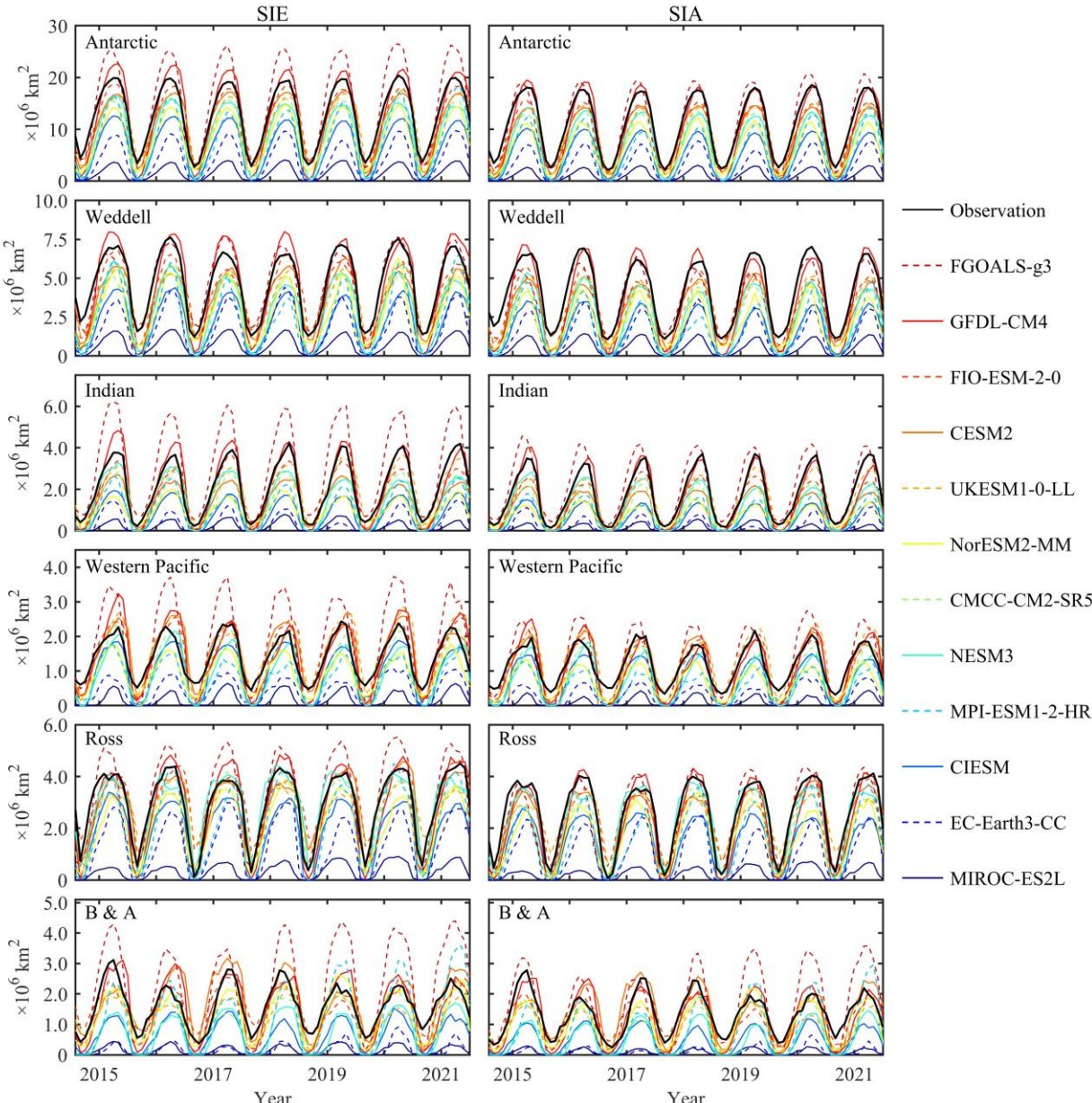

**Figure 2.** Monthly variation of sea ice extent (SIE) and sea ice area (SIA) for the observation and CMIP6 models in the whole Antarctic and five subregions.

In general, most models predict smaller mean SIE and SIA than the observation (Table 2). However, some exceptions mainly occur in subregions. For example, compared to the observation, FGOALS-g3 shows larger mean SIE in all subregions and larger mean SIA in the Indian Ocean, Western Pacific Ocean, and B & A Seas. Besides, CESM2 shows larger mean SIE in both the Western Pacific Ocean and B & A Seas, as well as larger mean SIA in the B & A Seas. GFDL-CM4 and FIO-ESM-2-0 have larger SIE in the Western Pacific Ocean and Ross Sea, respectively. UKESM1-0-LL shows larger SIE and SIA in the Western Pacific Ocean. Considering the biases with the observation, FGOALS-g3 and GFDL-CM4 show the smallest differences of both SIE and SIA in the Weddell Sea and Indian Ocean, respectively. FIO-ESM-2-0 has the smallest biases of SIE in the Western Pacific Ocean and B & A Seas, as well as the smallest bias of SIA in the Ross Sea. CESM2 exhibits the smallest bias of SIE in the Ross Sea and the smallest bias of SIA in the B & A Seas. The smallest bias of SIA in the Western Pacific Ocean is from UKESM1-0-LL.

**Table 2.** Mean SIE/SIA over the period 2015–2021 for the observation and CMIP6 models in the whole Antarctic and five subregions.

| Name | Antarctic (×10⁶ km²) | Weddell (×10⁶ km²) | Indian (×10⁶ km²) | Western Pacific (×10⁶ km²) | Ross (×10⁶ km²) | B & A (×10⁶ km²) |
|---|---|---|---|---|---|---|
| Observation | 12.68/10.78 | 4.63/4.05 | 2.09/1.73 | 1.42/1.15 | 2.94/2.53 | 1.60/1.32 |
| CESM2 | 11.15/8.69 | 3.60/2.85 | 1.39/0.97 | 1.53/1.04 | 2.91/2.22 | 1.72/1.34 |
| CIESM | 6.48/4.79 | 2.15/1.60 | 0.82/0.58 | 0.97/0.71 | 1.87/1.40 | 0.67/0.49 |
| CMCC-CM2-SR5 | 9.12/7.02 | 2.99/2.36 | 1.26/0.91 | 1.07/0.77 | 2.52/1.97 | 1.29/1.00 |
| EC-Earth3-CC | 4.10/2.97 | 1.76/1.30 | 0.39/0.25 | 0.44/0.30 | 1.34/1.02 | 0.17/0.10 |
| FGOALS-g3 | 14.85/10.65 | 4.65/3.52 | 3.05/2.09 | 1.85/1.25 | 3.13/2.25 | 2.17/1.55 |
| FIO-ESM-2-0 | 11.94/9.15 | 4.06/3.18 | 1.72/1.27 | 1.40/1.02 | 3.16/2.45 | 1.60/1.23 |
| GFDL-CM4 | 12.23/9.48 | 4.42/3.48 | 2.08/1.56 | 1.44/1.06 | 2.76/2.22 | 1.53/1.16 |
| MIROC-ES2L | 1.61/1.09 | 0.72/0.53 | 0.21/0.13 | 0.18/0.10 | 0.35/0.24 | 0.16/0.09 |
| MPI-ESM1-2-HR | 7.59/5.27 | 2.43/1.66 | 1.05/0.70 | 0.79/0.53 | 2.10/1.53 | 1.22/0.84 |
| NESM3 | 8.98/7.08 | 2.97/2.39 | 1.48/1.13 | 0.92/0.70 | 2.89/2.34 | 0.72/0.51 |
| NorESM2-MM | 8.15/6.05 | 2.84/2.05 | 0.99/0.72 | 0.93/0.66 | 2.10/1.64 | 1.30/0.98 |
| UKESM1-0-LL | 11.22/9.04 | 3.74/3.09 | 1.92/1.49 | 1.59/1.24 | 2.76/2.28 | 1.20/0.94 |

Over the period 2015–2021, the twelve CMIP6 models show significant differences in variation trends with the observation based on SIE and SIA anomalies (Figure 3). The observed SIE and SIA only have slightly negative trends (less than $-0.01 \times 10^6$ km²·year⁻¹), while the models show the strong negative trends from $-0.33 \times 10^6$ km²·year⁻¹ to $-2.74 \times 10^6$ km²·year⁻¹ for SIE and from $-0.23 \times 10^6$ km²·year⁻¹ to $-2.36 \times 10^6$ km²·year⁻¹ for SIA. MIROC-ES2L has the smallest variation trend among the twelve models, and the trend is the most similar to the observed trend. In addition, its SIE and SIA are significantly lower than the observation (Figure 2). All twelve CMIP6 models overestimated the negative trends of SIE and SIA. These strong negative trends of SIE and SIA were also found in other periods, such as during 1979–2014 [19] or 1979–2018 [22].

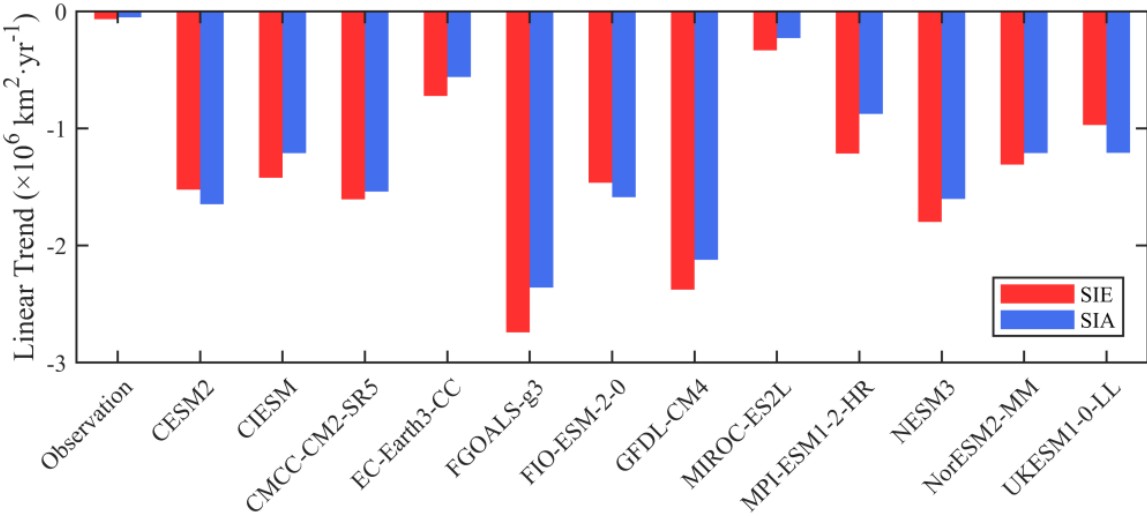

**Figure 3.** Comparisons of linear trend of SIE and SIA between the CMIP6 models and the observation in the whole Antarctic over the period 2015–2021.

Through the evaluation of RMSD, most models show a relatively small RMSD for SIE and SIA with the observation. In the whole Antarctic, the number of models with RMSD in SIE and SIA less than 20% of the observed mean SIE and SIA is nine and ten, respectively. The RMSD in the subregions is generally smaller than that of the whole Antarctic. CMCC-CM4-SR5 shows the smallest RMSD of SIE in the Antarctic ($0.98 \times 10^6$ km²) and the Indian Ocean ($0.35 \times 10^6$ km²). NESM3, NorESM2-MM, CIESM, and UKESM1-0-LL have the smallest RMSD in the Weddell Sea, Western Pacific Ocean, Ross Sea, and B & A Seas

(Figure 4). For SIA, CMCC-CM4-SR5, NESM3, CIESM, and UKESM1-0-LL still show the smallest RMSD in the Antarctic, Weddell Sea, Western Pacific Ocean, and B & A Seas (Figure 5). GFDL-CM4 and CESM2 have the smallest RMSD in the Indian Ocean and Ross Sea, respectively.

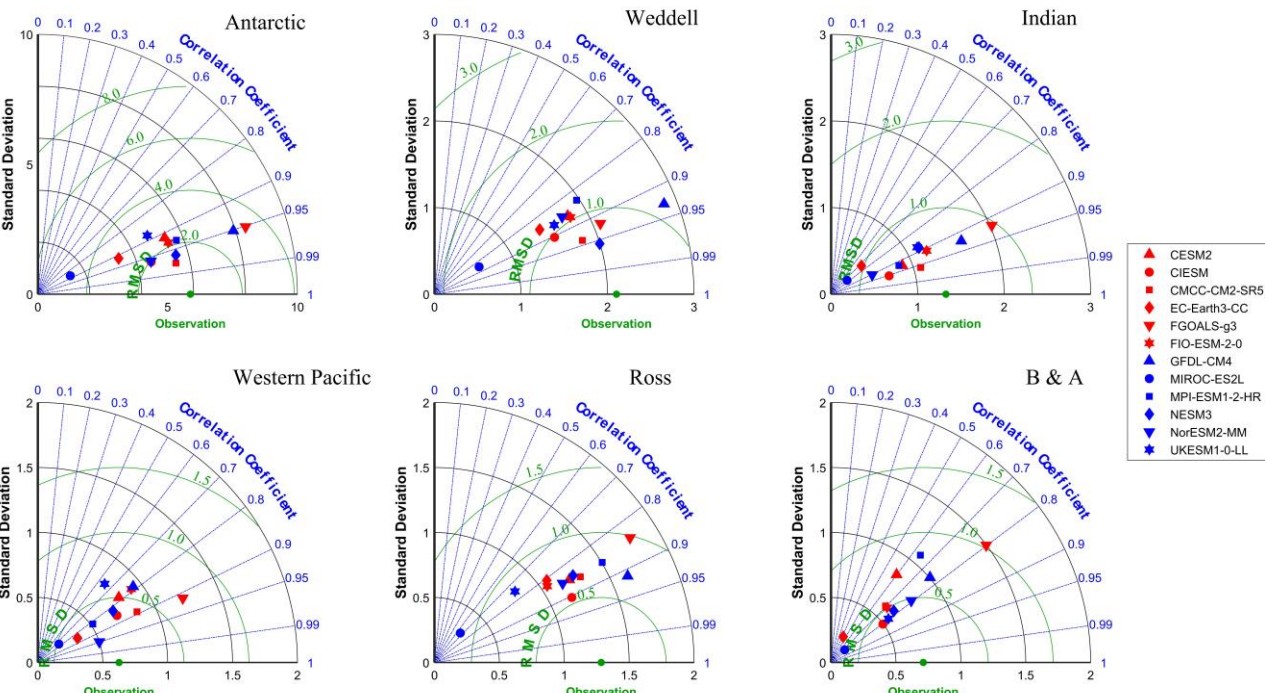

**Figure 4.** Taylor diagrams of SIE between the CMIP6 models and the observation in the whole Antarctic and five subregions over the period 2015–2021.

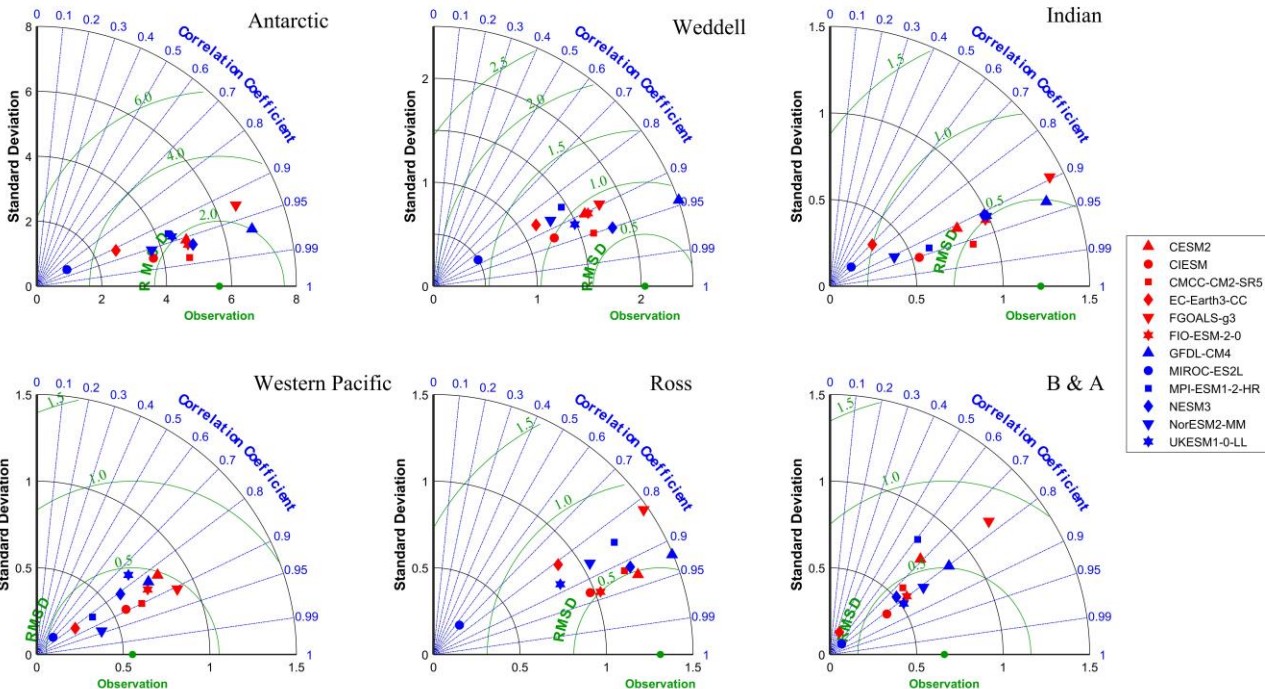

**Figure 5.** Taylor diagrams of SIA between the CMIP6 models and the observation in the whole Antarctic and five subregions over the period 2015–2021.

### 3.1.2. Seasonal Cycles

The seasonal cycle predictions of SIE and SIA are also evaluated. The differences between the models and the observation in the Antarctic indicate that both SIE and SIA show larger biases from May to December, especially in winter and spring (Figure 6). Compared to the observed mean SIE of $1.27 \times 10^7$ km$^2$ and SIA of $1.08 \times 10^7$ km$^2$, most models in CMIP6 underestimate the seasonal Antarctic SIE and SIA. MIROC-ES2L and EC-Earth3-CC show the two largest negative biases for both SIE and SIA with the mean value over $-10 \times 10^6$ km$^2$ in winter and spring. Among the models that underestimate SIE and SIA throughout the year, CMCC-CM2-SR5 has the smallest mean bias of SIE ($-3.55 \times 10^6$ km$^2$), while NESM3 has the smallest mean bias of SIA ($-3.70 \times 10^6$ km$^2$).

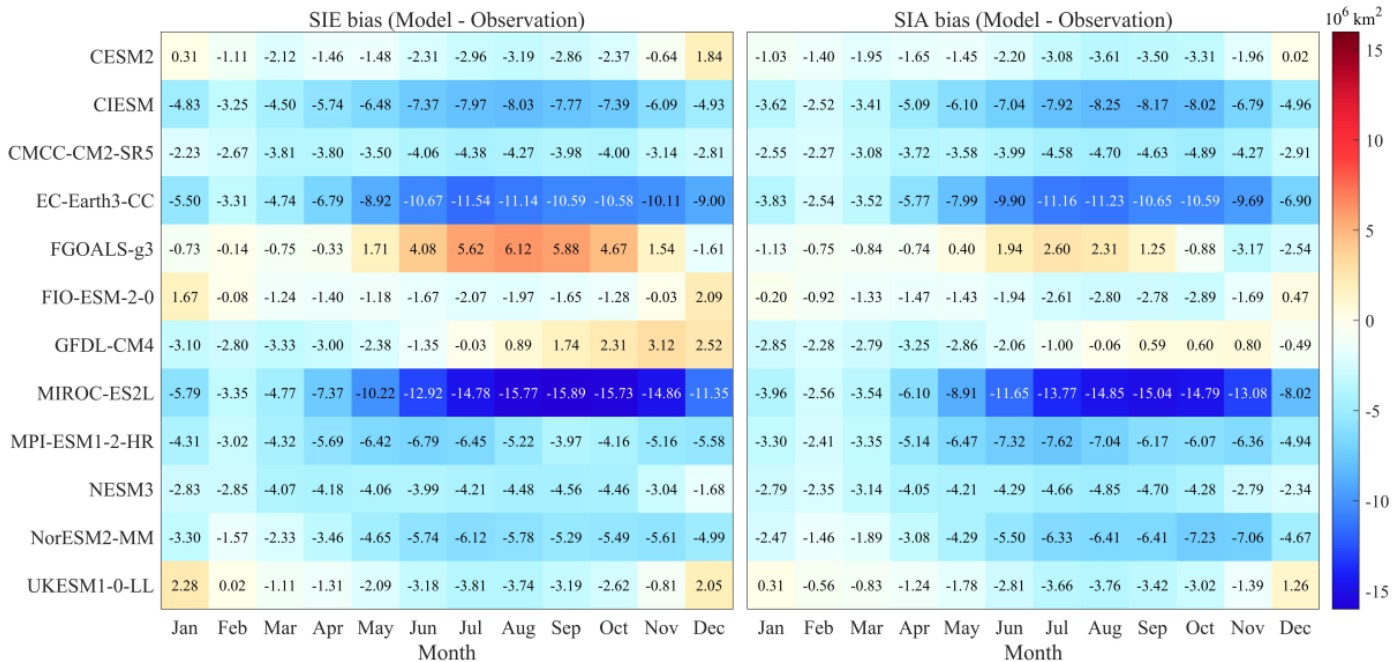

**Figure 6.** Seasonal cycle of SIE (**left**) and SIA (**right**) biases between the CMIP6 models and the observation during 2015–2021.

Five models exhibit overestimation of SIE and SIA in some months. FGOALS-g3 shows the largest mean positive biases of $4.23 \times 10^6$ km$^2$ for SIE from May to December and $1.70 \times 10^6$ km$^2$ for SIA from May to September, with the peak positive bias in SIE and SIA appearing in August and July, respectively. GFDL-CM4 overestimates SIE from August to December. Three additional models (CESM2, FIO-ESM-2-0, and UKESM1-0-LL) mainly show positive biases of SIE in December and January. For SIA, these positive biases are reduced, and some of them even decline to negative biases. In the twelve models, the offset of positive and negative biases results in the smallest mean bias of SIE and SIA from GFDL-CM4 and FGOALS-g3, respectively, with the values of $-4.50$ and $-1.21 \times 10^5$ km$^2$.

### 3.1.3. Interannual Variation

The interquartile ranges of SIE and SIA generated by using the annual mean observed data indicate significant interannual fluctuations over the period 2015–2021 (Figure 7). With the exception of CIESM, MPI-ESM1-2-HR, and UKESM1-0-LL, most of the CMIP6 model prediction results show smaller interannual variations than the observation. The model with the largest interannual fluctuation is MPI-ESM1-2-HR, with $1.41 \times 10^6$ km$^2$ for SIE and $1.08 \times 10^6$ km$^2$ for SIA. The extremely small interannual variations are presented by CMCC-CM2-SR5 and MIROC-ES2L. In addition, most models are able to capture the observed characteristics that the interquartile range of SIE is larger than SIA with the exception of

NorESM2-MM, which suggests larger interannual fluctuations of SIE. Moreover, we found that the interannual fluctuations in the models are smaller than the intermodel differences.

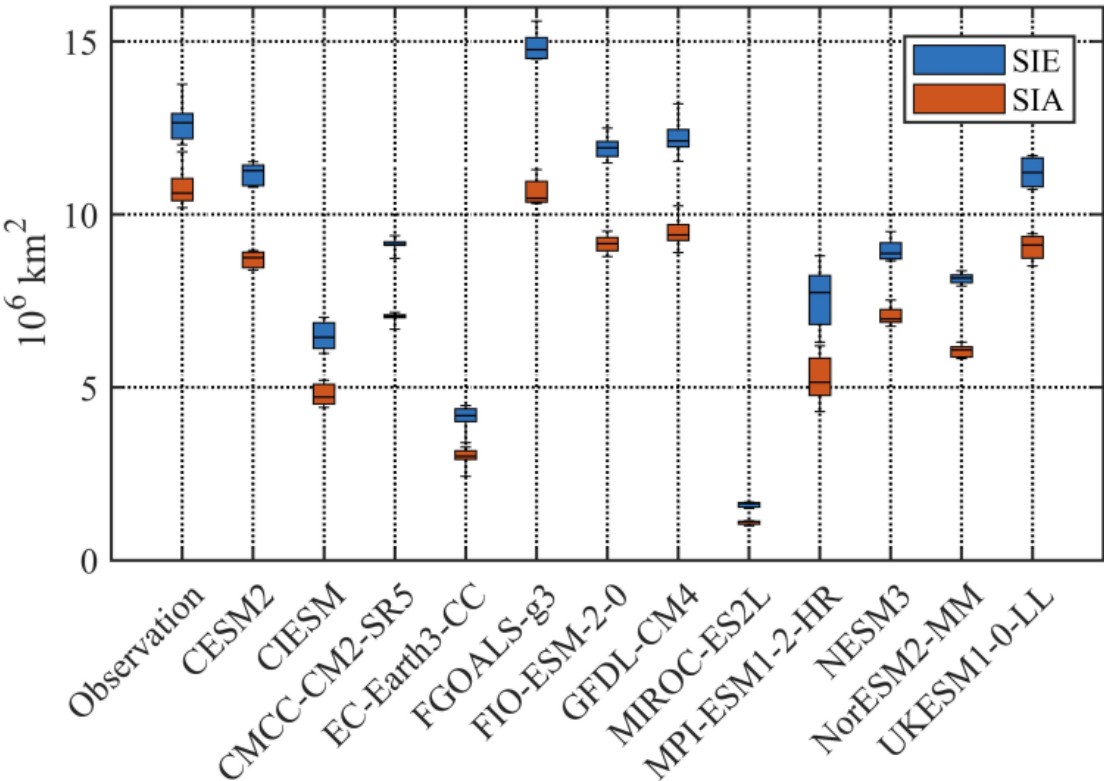

**Figure 7.** Interannual comparison of SIE and SIA between the CMIP6 models and the observation. The box covers the inter-quartile range with the solid line inside the box representing the median value. Whiskers denote the maximum and minimum values.

The medians of SIE and SIA from the models are mostly smaller than those observed, and the interquartile ranges from the models are generally smaller than the interquartile ranges from the observation. This tendency indicates that most models have underestimated the interannual variation of SIE and SIA. Based on the comprehensive analysis of the median and interquartile range, we consider GFDL-CM4 and FIO-ESM-2-0 to have more reasonable interannual agreements with the observation for SIE and SIA.

### 3.2. The Assessment of SIC Prediction

3.2.1. Spatial Distribution of SIC

Besides SIE and SIA, the predictions of SIC in CMIP6 models are also evaluated in detail in this study. The spatial distribution of observed SIC shows a major pattern that larger SIC (over 95%) are located in the coastal regions surrounding the Antarctic continent (Figure 8). Most models can capture the large SIC surrounding the coastal regions of Antarctica, and some models such as CESM2, CMCC-CM2-SR5, EC-Earth3-CC, FGOALS-g3, GFDL-CM4, NorESM2-MM, and UKESM1-0-LL are able to predict the larger SIC in the western Weddell Sea. However, most models fail to accurately reproduce the distribution of SIC near the sea ice edge. These spatial differences of SIC in the CMIP6 models result in significant discrepancies of SIE and SIA.

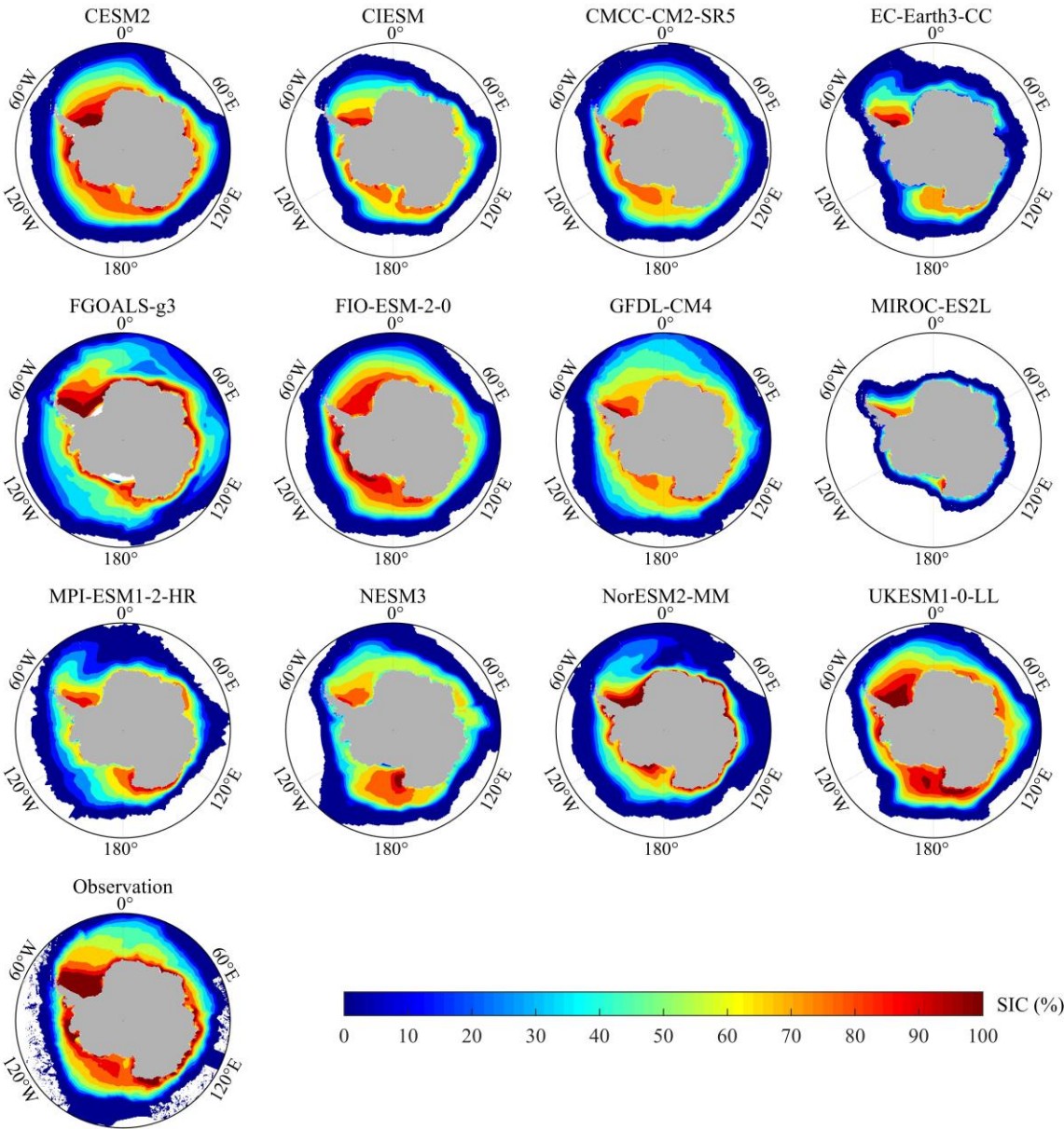

**Figure 8.** Spatial distributions of multi-year mean SIC for the CMIP6 models and the observation over the period 2015–2021. Areas without sea ice are filled by blank space.

To conduct a detailed comparison between the CMIP6 models and the observation, we selected the SIC in February and September, the most representative months with the minimum and maximum SIC. Our analysis reveals that, in February, the CMIP6 models exhibit negative biases primarily concentrated in the coastal regions surrounding the Antarctic continent. Among them, FGOALS-g3 and UKESM1-0-LL exhibit relatively better predictions, as their mean SIC differences were less than −10% in the western Weddell Sea (Figure 9). In addition, some models, such as FIO-ESM-2-0, NESM3, NorESM2-MM, and UKESM1-0-LL in the Ross Sea and FGOALS-g3 in the Indian Ocean have positive biases in some subregions obviously. Similar to the tendency in February, the SIC biases in September are still dominated by negative values (Figure 10). It is worth noting that the differences become more pronounced around the sea ice edge, rather than in regions with high SIC. FGOALS-g3 overestimated SIC significantly with a wide range of positive biases around the sea ice edge. GFDL-CM4 exhibits an overestimation of SIC in most regions of sea ice edge, while MIROC-ES2L exhibits a significant underestimation of SIC. On the other

hand, FIO-ESM-2-0 and GFDL-CM4 generally show smaller mean SIC bias of −8.1% and 2.8%, respectively, in September.

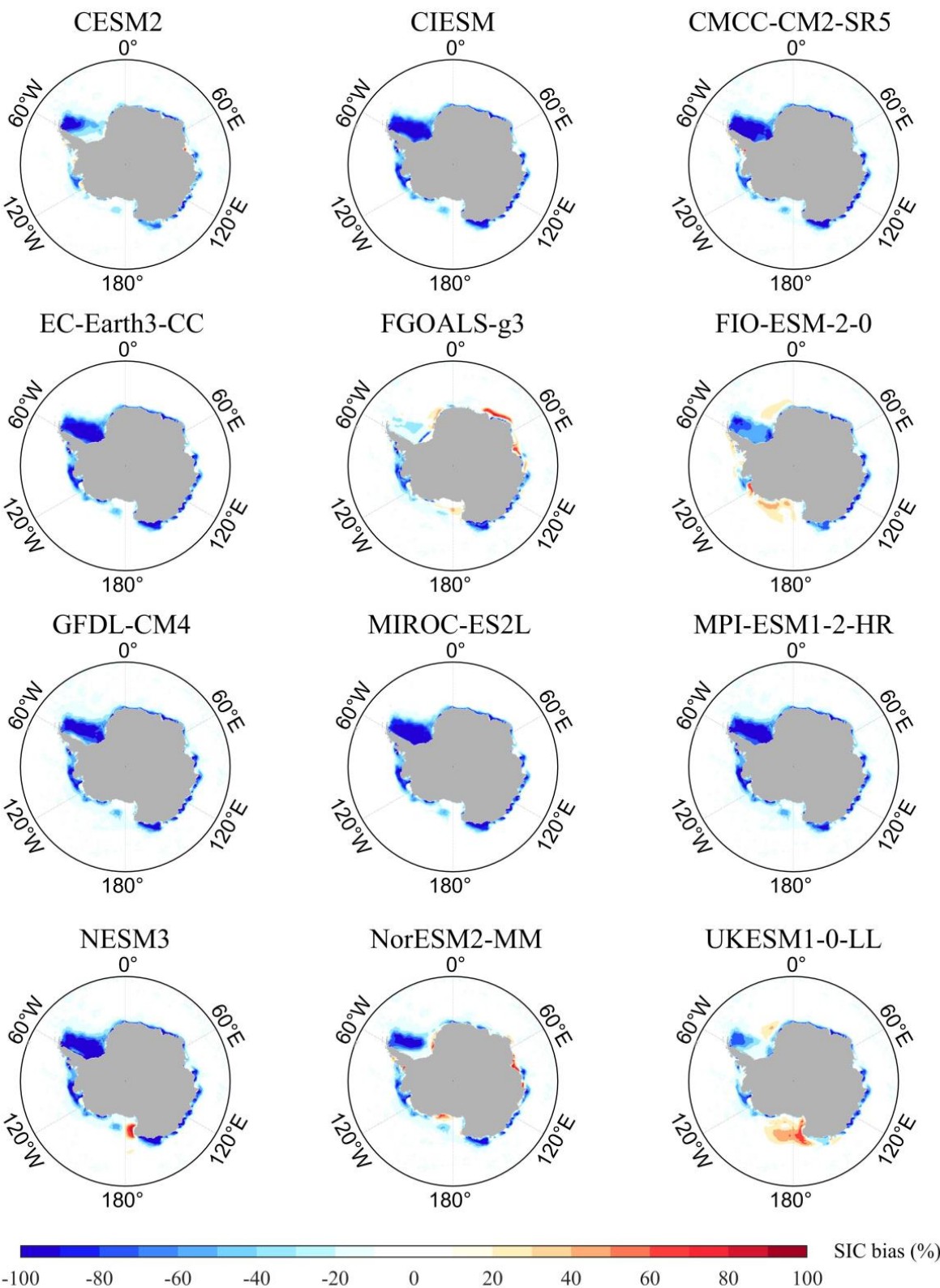

**Figure 9.** Spatial distributions of multi-year mean SIC difference in February between the CMIP6 models and the observation over the period 2015−2021.

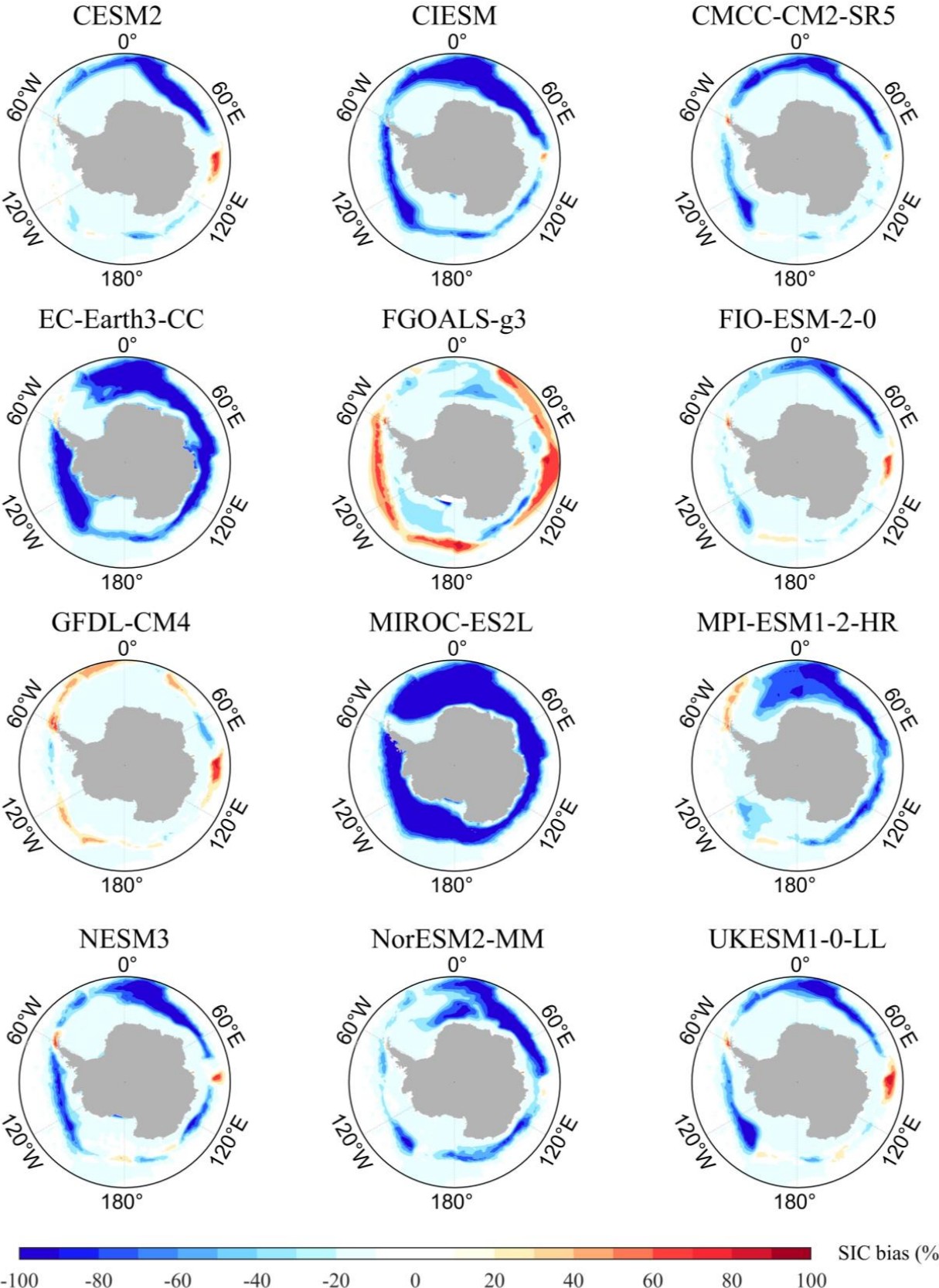

**Figure 10.** Spatial distributions of multi-year mean SIC difference in September between the CMIP6 models and the observation over the period 2015–2021.

According to the percentage statistics of SIC bias for all the data points over the period 2015–2021, almost all CMIP6 models show smaller SIC predictions (Figure 11). The comprehensive results indicate that five models (CESM2, FGOALS-g3, FIO-ESM-2-0, GFDL-CM4, and UKESM1-0-LL) have more reasonable results with the mean bias less than ±6% and the RMSD less than 27%. FGOALS-g3 has the smallest mean bias of 0.49% and is the only model showing the positive mean bias. GFDL-CM4 and FIO-ESM-2-0 show the two smallest RMSD (less than 23%) and negative mean bias (less than −5%). The biases less than ±10% account for over 60% of the total data points in GFDL-CM4, NESM3, and UKESM1-0-LL. MIROC-ES2L and EC-Earth3-CC exhibit larger percentages (over 30–40%) of negative SIC biases exceeding −90%, suggesting the two models predict extremely smaller SIE and SIA as already shown in Figure 2.

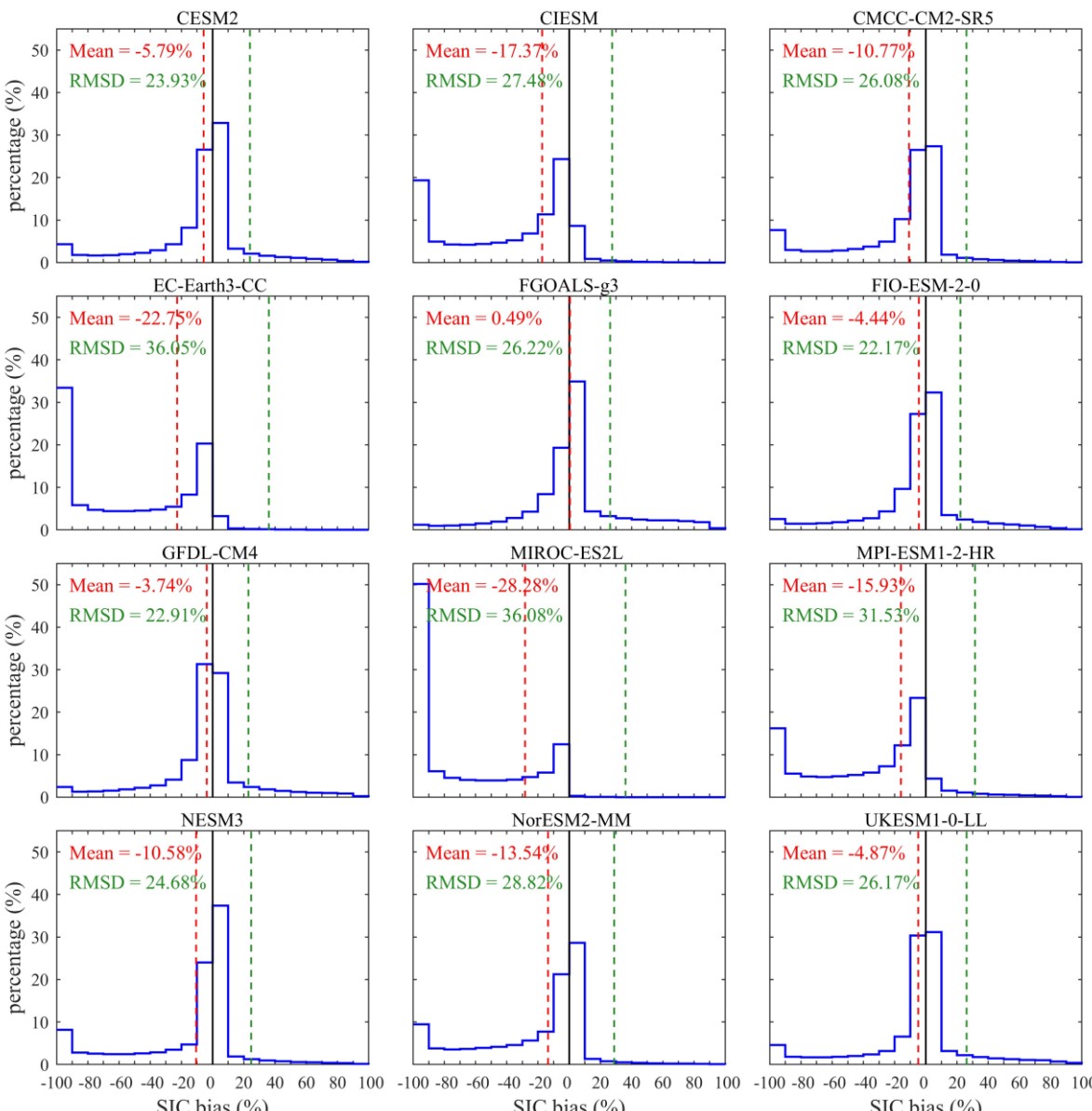

**Figure 11.** Percentage of SIC biases between the CMIP6 models and the observation for all the data points over the period 2015–2021. The solid black line indicates the boundary between negative and positive SIC bias. The red dashed line indicates the mean bias and the green dashed line indicates the RMSD.

### 3.2.2. Temporal Variation of SIC

Based on the correlation results between the CMIP6 models and the observation during 2015–2021, the majority of the most models exhibit reasonable agreement with the monthly variation of SIC in most parts of the Antarctic region (Figure 12). All models exhibit significant positive correlation coefficients ($p < 0.05$) covering more than 80% of the area. The highest mean positive correlation coefficient is 0.65 from GFDL-CM4. FIO-ESM-2-0, and CESM2 show correlation coefficients over 0.60 that are similar to GFDL-CM4. The consistency among these models is that they predict SIC variations more accurately in most coastal regions but struggle to reproduce SIC variations in some regions of the western Weddell Sea and the sea ice edge.

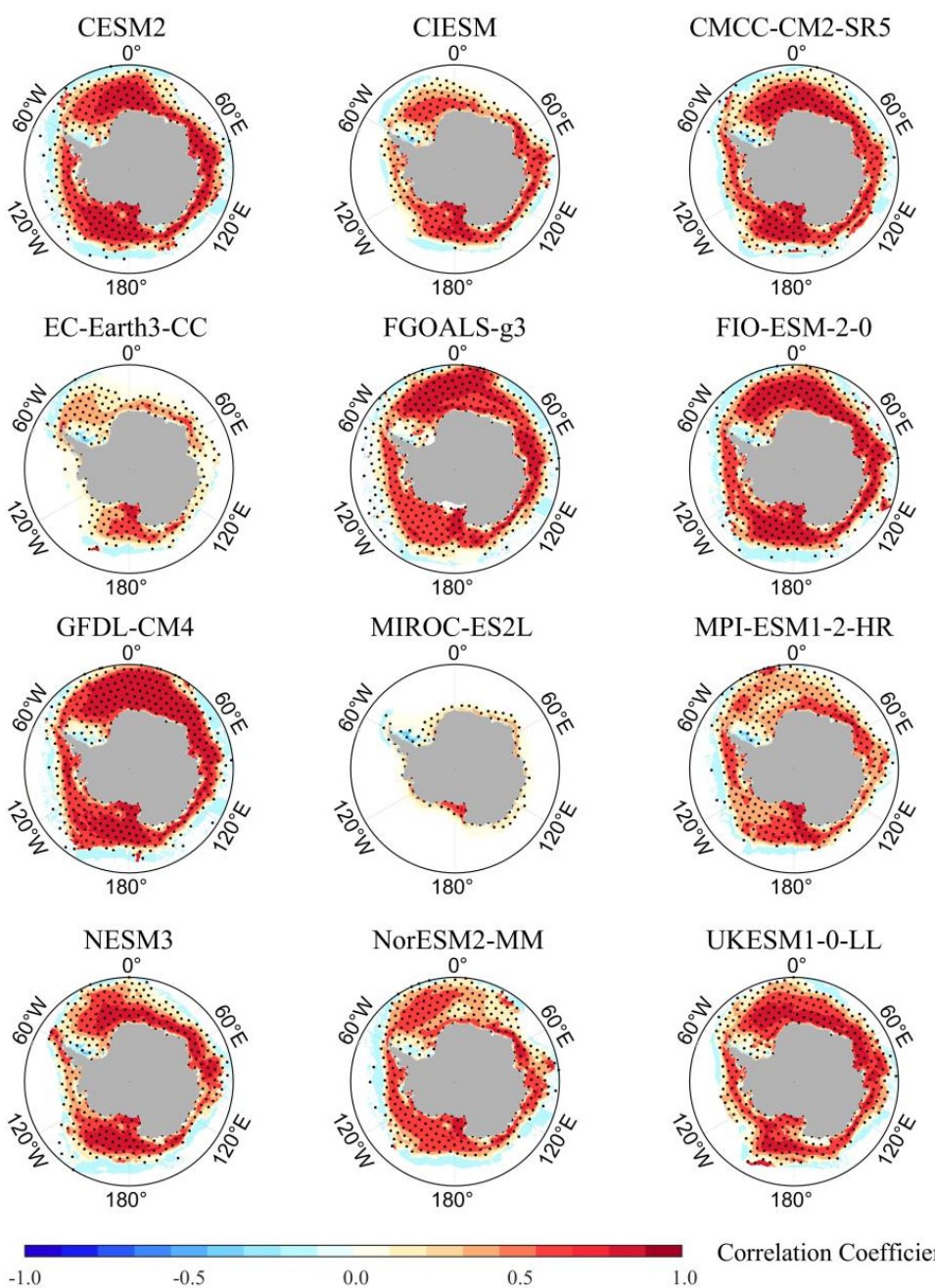

**Figure 12.** Distributions of correlation coefficients of SIC between the CMIP6 models and the observation. The black dots indicate the significance of the correlation coefficients at the 95% confidence level.

The observed SIC trends over the period 2015–2021 indicate a dominance of decreasing trends in the western Weddell Sea and eastern B & A Seas, and obvious increasing trends in the eastern Weddell Sea, the Indian Ocean, the Ross Sea, and the B & A Seas (Figure 13). Actually, few models are capable of accurately replicating these patterns. Compared to other models, CESM2, GFDL-CM4, and FGOALS-g3 partly exhibit more similar decreasing trends as observed. Both CESM2 and GFDL-CM4 have similar decreasing trends in the eastern B & A Seas, while FGOALS-g3 shows similarity to the observed trend in the Weddell Sea. CMCC-CM2-SR5, GFDL-CM4, and NESM3 display similarity to the observed increasing trend in the Ross Sea and B & A Seas. UKESM1-0-LL shows mainly similar increasing trends in the Indian Ocean.

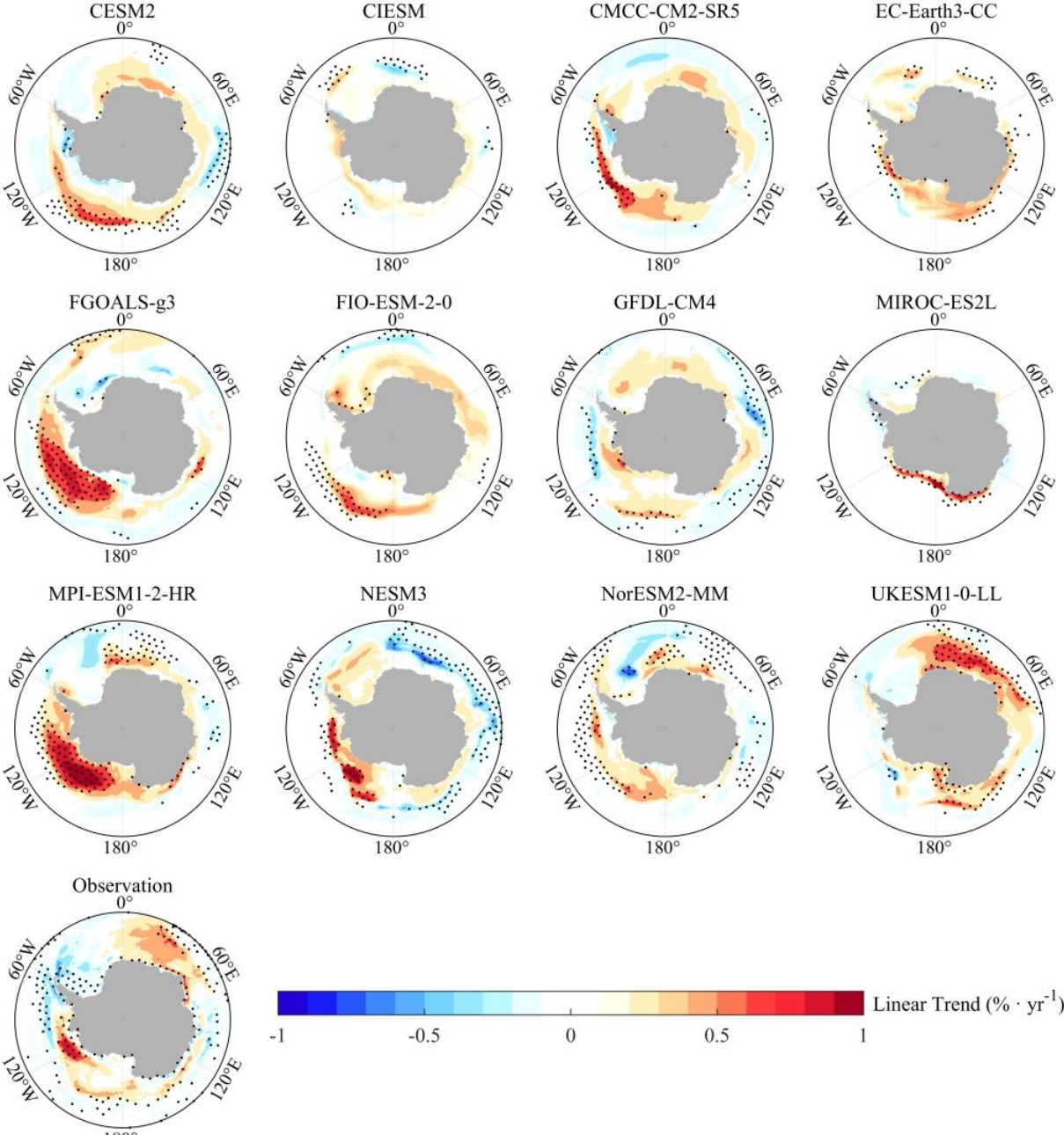

**Figure 13.** Linear trends of SIC for the CMIP6 models and the observation. The black dots indicate the regression slopes at the 95% confidence level.

## 4. Discussion

This study provides a comprehensive assessment of Antarctic sea ice cover prediction in CMIP6 climate models using statistical indicators such as correlation coefficient, bias, RMSD, median value, inter-quartile range, and linear trend. Among these statistical indicators, RMSD and correlation coefficient are commonly used to measure the accuracy of spatial differences and temporal variations, respectively, between models and observation. In fact, there exists no singular criterion to evaluate the performance of models. When the RMSD and correlation coefficient present inconsistent results, the prediction of Antarctic sea ice becomes challenging to verify. Therefore, additional metrics and evaluations are necessary. Here, we further employed the Taylor score (TS) to evaluate the SIE, SIA, and SIC. The TS metric comprehensively considers both the correlation coefficient and standard deviation, which had been effectively applied and verified in the evaluation of Arctic sea ice simulations [29]. In addition, we compare the predictive performance of sea ice cover using both RMSD and TS metrics. The resulting assessments are then scrutinized and discussed to determine differences and similarities.

For twelve models, TS in the subregions are generally lower than those in the whole Antarctic. Furthermore, TS for SIC is generally lower than TS of SIE and SIA. In the whole Antarctic, CESM2 and FIO-ESM-2-0 show superior performance, with TS over 0.90 for SIE, SIA, and SIC (Figure 14). Only CESM2 and FIO-ESM-2-0 consistently achieve TS over 0.60 for SIE, SIA, and SIC across all subregions. The TS of CMCC-CM2-SR5 is just slightly below 0.60 for SIC in both the Weddell Sea and B & A Seas.

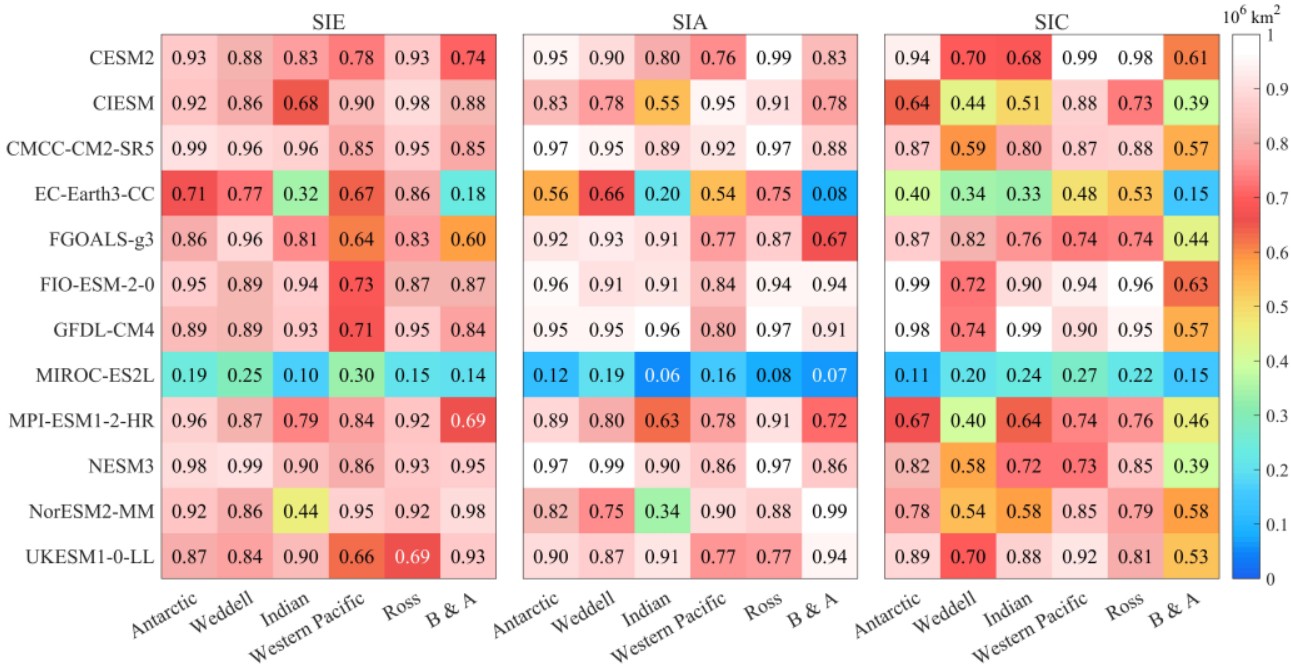

**Figure 14.** Taylor score of SIE, SIA, and SIC between the CMIP6 models and the observation in the whole Antarctic and five subregions over the period 2015–2021.

In addition, NESM3 also displays noteworthy performance, achieving TS over 0.90 for SIE and SIA in the whole Antarctic and over 0.85 in all subregions. Nevertheless, it falls short in terms of SIC with two lower TS values in the Weddell Sea (0.58) and B & A Seas (0.39). The TS of GFDL-CM4 is generally over 0.80 in most subregions, with the exceptions being SIE in the Western Pacific Ocean and SIC in the Weddell Sea and B & A Seas. Several models exhibit notable differences in TS for SIE, SIA, and SIC. For instance, CIESM, MPI-ESM1-2-HR, and NorESM2-MM perform better in terms of SIE but exhibit lower TS values for SIA and SIC. FGOALS-g3 displays the higher TS for SIE and SIA but has the lower TS for SIC.

By comparing the results obtained through both RMSD and TS, Table 3 indicates that the two assessment methods produce almost consistent results in all regions, highlighting the applicability of TS in the assessment of Antarctic sea ice, with the results being representative. The only exception pertains to the B & A Seas, where UKESM1-0-LL performed best for SIE and SIA based on RMSD, while Nor-ESM2-MM had the best performance for SIE and SIA based on TS. In the Antarctic, CMCC-CM2-SR5 has the most reasonable prediction results for both SIE and SIA. In contrast, in terms of predicting SIC, FIO-ESM-2-0 outperforms other models. In the assessment of SIE and SIA, the best prediction performance of models varies across different subregions. However, CESM2 performs most reasonably in terms of predicting SIC in three subregions.

**Table 3.** Comparison of the most reasonable prediction of SIE, SIA, and SIC based on RMSD and TS in the whole Antarctic and five subregions.

| Region | SIE | | SIA | | SIC | |
|---|---|---|---|---|---|---|
| | RMSD | TS | RMSD | TS | RMSD | TS |
| Antarctic | CMCC-CM2-SR5 | CMCC-CM2-SR5 | CMCC-CM2-SR5 | CMCC-CM2-SR5 /NESM3 | FIO-ESM-2-0 | FIO-ESM-2-0 |
| Weddell | NESM3 | NESM3 | NESM3 | NESM3 | FGOALS-g3 | FGOALS-g3 |
| Indian | CMCC-CM2-SR5 | CMCC-CM2-SR5 | GFDL-CM4 | GFDL-CM4 | GFDL-CM4 | GFDL-CM4 |
| Western Pacific | Nor-ESM2-MM | Nor-ESM2-MM | CIESM | CIESM | CESM2 | CESM2 |
| Ross | CIESM | CIESM | CESM2 | CESM2 | CESM2 | CESM2 |
| B & A | UKESM1-0-LL | Nor-ESM2-MM | UKESM1-0-LL | Nor-ESM2-MM | CESM2 | CESM2 |

## 5. Conclusions

Sea ice plays a significant role in regulating air–sea interactions and can significantly impact the generation and propagation of ocean waves. In the context of recent global warming, the ability to make reasonable predictions regarding Antarctic sea ice is essential in advancing our understanding of the variations in Antarctic sea ice and its response to future climate change. In this study, a comprehensive assessment of the Antarctic sea ice cover prediction has been conducted using twelve CMIP6 models under the SSP2-4.5 scenario, by means of comparing with the satellite observed data of AMSR2. The performances of SIE, SIA, and SIC prediction in these models are evaluated in detail based on the quantification of spatio-temporal differences.

In the assessments of SIE and SIA, most CMIP6 models can show reasonable monthly variation of SIE with the observation. Specifically, CMCC-CM4-SR5 has the highest correlation coefficient of 0.98 in the Antarctic and successfully captures the minima in February and maxima in September. In the subregions, the highest correlation of SIE is exhibited by various models. Compared to SIE, the CMIP6 models show relatively higher correlations of SIA. CMCC-CM4-SR5 shows the largest correlation coefficient of SIA in the whole Antarctic. Moreover, the largest correlation coefficients in the subregions are still observed in different models. However, none of the models accurately capture the time periods of minima and maxima in all subregions. It is notable that all the twelve models overestimated the negative trends of SIE and SIA.

Most models show relatively small RMSD for SIE and SIA with the observation. The RMSD values in the whole Antarctic are generally smaller than those in the subregions. Among the models, CMCC-CM4-SR5 still displayed the best performance with the smallest RMSD for SIE and SIA. However, the models with the smallest RMSD of SIE and SIA in the subregions are inconsistent.

In general, most models predict smaller mean SIE and SIA than the observation, and the smallest mean bias of SIE and SIA is from GFDL-CM4 and FGOALS-g3, respectively, with the values of $-4.50$ and $-1.21 \times 10^5$ km$^2$. In the seasonal cycle comparison, seven models underestimate SIE and SIA in all the months. Among these models, CMCC-

CM2-SR5 and NESM3 have the smallest mean bias of SIE ($-3.55 \times 10^6$ km$^2$) and SIA ($-3.70 \times 10^6$ km$^2$).

In addition, the analysis of the median and interquartile range reveals that the inter-annual variation of SIE and SIA in most CMIP6 models is smaller than the observation. GFDL-CM4 and FGOALS-g3 have the most reasonable interannual agreement with the observation for SIE and SIA, respectively.

In the assessments of SIC, most models show large SIC surrounding the Antarctic coastal regions, but few models can accurately reproduce all the major distribution patterns, especially in the region of sea ice edge. Eleven CMIP6 models underestimate SIC, while five models (CESM2, FIO-ESM-2-0, GFDL-CM4, FGOALS-g3, and UKESM1-0-LL) have relatively small biases and RMSD values, which are less than $\pm6\%$ and 27%, respectively. Among these five models, FGOALS-g3 shows the smallest mean bias of 0.49%, while GFDL-CM4 and FIO-ESM-2-0 show relatively small RMSD (less than 23%) and negative mean bias (less than $-5\%$). Furthermore, the correlation of SIC variation between the models and the observation in the Antarctic region is generally low. However, GFDL-CM4, FIO-ESM-2-0, and CESM2 exhibit larger positive correlation coefficients over 0.60. Most models struggle to accurately replicate the distribution patterns of variation trends, and GFDL-CM4 shows more similarity in capturing some major patterns of both increasing and decreasing trends.

Another statistical indicator, Taylor score (TS), is included in this study for the assessment of SIE, SIA, and SIC. The assessment results between TS and RMSD are compared and discussed. Only two models, CESM2 and FIO-ESM-2-0, show TS over 0.90 in the Antarctic and over 0.60 in all subregions for SIE, SIA, and SIC. TS shows almost consistent results with RMSD in evaluating the most reasonable sea ice cover prediction in the whole Antarctic and its subregions, suggesting its suitability as a metric for assessing Antarctic sea ice cover, with the evaluation results being indicative.

Through the comprehensive assessments of sea ice cover above, CESM2, CMCC-CM4-SR5, FGOALS-g3, FIO-ESM-2-0, and GFDL-CM4 provide more reasonable predictions of sea ice cover over the period 2015–2021. The findings are significant in aiding the comprehension of the uncertainties associated with the CMIP6 models and facilitating the future selection of a multimodel ensemble. It is evident that additional endeavors to assess more CMIP6 models and for a longer duration are imperative to enhance the ability to predict Antarctic sea ice. This will facilitate the refinement and improvement of the models' predictive capabilities, enabling more accurate projections of sea ice cover in the future.

**Author Contributions:** Conceptualization, Y.Z. (Yu Zhang) and C.C.; methodology, S.L. and Y.Z. (Yu Zhang); validation, S.L., Y.Z. (Yu Zhang), and S.H.; formal analysis, S.L.; data curation, Y.Z. (Yu Zhang); writing—original draft preparation, S.L. and Y.Z. (Yu Zhang); writing—review and editing, C.C., Y.Z. (Yiran Zhang), D.X., and S.H.; visualization, S.L. All authors have read and agreed to the published version of the manuscript.

**Funding:** This research was funded by National Natural Science Foundation of China (No. 42130402), Innovation Group Project of Southern Marine Science and Engineering Guangdong Laboratory (Zhuhai) (No. 311022006), and the National Key Research and Development Program of China (No. 2018YFC14106801).

**Data Availability Statement:** The CMIP6 data of sea ice concentration is downloaded from https://esgf-node.llnl.gov/search/cmip6, accessed on 6 August 2021. The AMSR2 data of sea ice concentration is available at https://nsidc.org/data/au_si12/versions/1, accessed on 28 November 2022.

**Acknowledgments:** We would like to thank the editors and the anonymous reviewers for their constructive comments and advice.

**Conflicts of Interest:** The authors declare no conflict of interest.

## Abbreviations

The following abbreviations about the institutions are used in this manuscript:

| | |
|---|---|
| NCAR | National Center for Atmospheric Research |
| THU | Tsinghua University |
| CMCC | Euro-Mediterranean Center on Climate Change |
| EC | European Commission |
| FIO | First Institute of Oceanography, Ministry of Natural Resources |
| NOAA | National Oceanic and Atmospheric Administration |
| GFDL | Geophysical Fluid Dynamics Laboratory |
| CAS | Chinese Academy of Sciences |
| MIROC | Model for Interdisciplinary Research on Climate |
| NUIST | Nanjing University of Information Science and Technology |
| NCC | Norwegian Climate Center |
| MPI-M | Max Planck Institute for Meteorology |
| NCAS | National Centre for Atmospheric Science |
| MOHC | The Met Office Hadley Centre |

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
