# Peer review of "Assessment of Antarctic Sea Ice Cover in CMIP6 Prediction with Comparison to AMSR2 during 2015–2021"

_remotesensing, doi:10.3390/rs15082048_

Round 1

Reviewer 2 Report

The reviewer thanks the authors for a comprehensive and useful study comparing climate models and satellite data of Antarctic sea ice parameters. Its a thorough study, and the reviewer has no further comment on the data, methods, results and discussion other than the minor details as follows:

- The reviewer would like to see some discussion on why the 12 particular CMIP models were chosen.

- The technical details of the AMSR2 and why it is suited to sea ice imaging are important.

The major difficulty the reviewer has is that some of the English language turns of phrase made the paper difficult to read - and indeed, some sentences made no sense. The reviewer apologises for making such a comment because they have no other language, unlike the skill of these scholars. Nonetheless, it is essential that the paper receive some English language editing, please.

Reviewer 3 Report

Review on “Assessment of Antarctic Sea Ice Coverage in CMIP6 Prediction with Comparison to AMSR2 during 2015–2021”, by Siqi Li, Yu Zhang, Changsheng Chen, Yiran Zhang, Danya Xu and Song Hu, submitted for publication in Remote Sensing.

General comments :

The paper evaluates the performance of 12 global climate coupled models with respect to the Antarctic sea ice using retrievals of sea ice concentration from passive microwave observations (AMSR2), under the intermediate greenhouse gas emissions scenario (SSP2-4.5). Results indicate large variations among the models, and some models show large differences with respect to the observational dataset.

This study should be useful for the model developers to address some serious issues regarding sea ice around the Antarctic.

Only a few corrections are suggested in the specific comments section below, but a thorough review of the manuscript is required regarding the English language.

Specific comments:

Line 14-15: “…relatively smaller difference with…” à “…relatively small difference with…”

Line 98:girds” à “grids”

Line 120: “Σsmà “σsm

Round 2
